# Comparative chromosome painting in three Pelecaniformes species (Aves): Exploring the role of macro and microchromosome fusions in karyotypic evolution

**Igor Chamon Assumpção Seligmann[1], Ivanete de Oliveira Furo[2], Michelly da Silva dos Santos[3], Ricardo José Gunski[4], Analía del Valle Garnero[4], Fabio Augusto Oliveira Silva[5], Patricia O´Brien[6], Malcolm Ferguson-Smith[6], Rafael Kretschmer[7], Edivaldo Herculano C. de Oliveira[8,9]***

1 Programa de Pós-graduação em Biodiversidade e Biotecnologia da Rede Bionorte, Universidade Federal do Pará, Belém, Pará, Brazil, 2 Laboratório de Reprodução Animal, LABRAC, Universidade Federal Rural da Amazônia, UFRA, Parauapebas, State of Pará, Brazil, 3 Programa de Pós-graduação em Genética e Biologia Molecular, Universidade Federal do Pará, Belém, State of Pará, Brazil, 4 Programa de Pós-graduação em Ciências Biológicas, Universidade Federal do Pampa, Campus São Gabriel, São Gabriel, State of Rio Grande do Sul, Brazil, 5 Programa de Pós-graduação em Neurociência e Biologia Molecular, Universidade Federal do Pará, Belém, State of Pará, Brazil, 6 Cambridge Resource Centre for Comparative Genomics, University of Cambridge, Cambridge, United Kingdom, 7 Departamento de Ecologia, Zoologia e Genética, Universidade Federal de Pelotas, Pelotas, State of Rio Grande do Sul, Brazil, 8 Faculdade de Ciências Naturais, Instituto de Ciências Exatas e Naturais, Universidade Federal do Pará, Belém, State of Pará, Brazil, 9 Laboratório de Citogenômica e Mutagênese Ambiental, SEAMB, Instituto Evandro Chagas, Ananindeua, State of Pará, Brazil

* ehco@ufpa.br

**Data Availability Statement:** All relevant data are within the paper.

## Abstract

Pelecaniformes is an order of waterbirds that exhibit diverse and distinct morphologies. Ibis, heron, pelican, hammerkop, and shoebill are included within the order. Despite their fascinating features, the phylogenetic relationships among the families within Pelecaniformes remain uncertain and pose challenges due to their complex evolutionary history. Their karyotypic evolution is another little-known aspect. Therefore, to shed light on the chromosomal rearrangements that have occurred during the evolution of Pelecaniformes, we have used whole macrochromosome probes from *Gallus gallus* (GGA) to show homologies on three species with different diploid numbers, namely *Cochlearius cochlearius* (2n = 74), *Eudocimus ruber* (2n = 66), and *Syrigma sibilatrix* (2n = 62). A fusion between GGA6 and GGA7 was found in *C. cochlearius* and *S. sibilatrix*. In *S. sibilatrix* the GGA8, GGA9 and GGA10 hybridized to the long arms of biarmed macrochromosomes, indicating fusions with microchromosomes. In *E. ruber* the GGA7 and GGA8 hybridized to the same chromosome pair. After comparing our painting results with previously published data, we show that distinct chromosomal rearrangements have occurred in different Pelecaniformes lineages. Our study provides new insight into the evolutionary history of Pelecaniformes and the chromosomal changes involving their macrochromosomes and microchromosomes that have taken place in different species within this order.

**Funding:** This research was funded by Conselho Nacional de Desenvolvimento Científico e Tecnológico (CNPq, Proc. 407285/2021-0 to Analía del Valle Garnero, Proc. 304781/2022-3 to Edivaldo Herculano C. de Oliveira) and Pró-Reitoria de Pesquisa e Pós-Graduação – UFPA (Edital 02/2023-PAPQ to Edivaldo Herculano Corrêa de Oliveira). The funders had no role in study design, data collection and analysis, decision to publish, or preparation of the manuscript.

**Competing interests:** The authors have declared that no competing interests exist.

## Introduction

The order Pelecaniformes includes waterbirds such as ibis, herons, pelicans, hammerkop, and shoebill, totaling approximately 120 species [1]. The phylogenetic classification of this order has been much debated [2–7]. Currently, five families are included in the order Pelecaniformes: Threskiornithidae, Ardeidae, Scopidae, Balaenicipitidae and Pelecanidae, of which Ardeidae and Threskiornithidae are the most species-rich families [1]. Traditionally, most of these families (Threskiornithidae, Ardeidae, Scopidae and Balaenicipitidae) were included in the order Ciconiiformes. However, Ciconiiformes includes at present only the family Ciconiidae [1]. Concerning the phylogenetic proximity with other orders, Suliformes (waterbirds) are considered the sister group to the Pelecaniformes and Ciconiiformes are the sister group to the clade formed by Suliformes and Pelecaniformes [8]. The relationships among Pelecaniform bird families remain uncertain [3, 5].

Regarding the cytogenetic data, an interesting variation in diploid numbers (2n) has been described among Pelecaniformes species. To date, the lowest 2n is 50 in *Plegadis falcinellus* [9], the highest is 74 in *Cochlearius cochlearius* [10] and most species show 2n around 68 [11]. Hence, compared to the typical diploid number of birds, including the putative ancestral number close to 2n = 80 [12, 13], we can assume that Pelecaniformes includes species with derived karyotypes.

Chromosome painting has proved to be excellent in identifying syntenic chromosomes and chromosomal rearrangements [13], which have led to some interesting assumptions in phylogenetic and biogeography context of some groups [14–16]. Recently, a study investigated the chromosome organization using molecular cytogenetics in three Pelecaniformes species, *Ardea cinerea* and *Egretta garzetta* from the Ardeidae family, both with 2n = 64 and *Nipponia nippon* from the Threskiornithidae family, with 2n = 68 [17]. Comparative chromosome painting in these species using whole chromosome probes from the stone curlew (*Burhinus oedicnemus*, BOE, 2n = 42) and some from chicken (*Gallus gallus*, GGA, 2n = 78) indicated that distinct chromosome rearrangements had occurred in different Pelecaniformes lineages. These authors also inferred homologies with chicken from previous studies. The main rearrangements were fusion events, involving both macro- and micro-chromosomes. The authors proposed that the fusion between the chromosomes homologous to GGA 7 and 8 is a potential cytogenetic signature that unites Ardeidae and Threskiornithidae.

Molecular cytogenetics studies in orders closely related to Pelecaniformes are also scarce. Among the Suliformes, only *Phalacrocorax brasilianus* (2n = 74) has been investigated with chicken BACs probes [18]. This study revealed that the chromosome evolution in *P. brasilianus* included fissions of macrochromosomes, and fusions involving both macro and microchromosomes. Among the Ciconiiformes, only *Jabiru mycteria* (2n = 56) and *Ciconia maguari* (2n = 72) have been investigated with *G. gallus* and *Leucopternis albicollis* paints [19]. These authors found that the reduction in the diploid number of Ciconiiformes members were due to chromosome fusions occurring in both macro and microchromosomes.

Hence, we investigate here the chromosome organization in three Pelecaniformes species, *C. cochlearius* and *Syrigma sibilatrix* from the Ardeidae family, and *Eudocimus ruber* from the Threskiornithidae family. The species were selected because the *C. cochlearius* is considered phylogenetically a basal species in the family [20] and *S. sibilatrix* is more derived [21]. We used *G. gallus* chromosome probes to explore syntenies and chromosomal rearrangements. The results are used to discuss the chromosomal rearrangements in a genomic and phylogenetic context, comparing our results with previous data from Pelecaniformes, Ciconiiformes and Suliformes species.

## Materials and methods

### Animals and chromosome preparations

Three species belonging to two different families of Pelecaniformes were studied: *C. cochlearius* (CCO, one male) and *S. sibilatrix* (SSI, one female) from the Ardeidae family, and *E. ruber* (ERU, one male) from the Threskiornithidae family. The biological material was obtained following permissions from Sistema de Autorização e Informação em Biodiversidade (SISBIO 33860–3 and 44173–1—ICMBio). The experiments followed protocols approved by the ethics committee from Universidade Federal do Pampa (no. 026/2012). *C. cochlearius* and *E. ruber* were collected at Parque Zoológico Mangal das Garças, Pará State, Brazil, while *S. sibilatrix* was collected in São Gabriel, Rio Grande do Sul State, Brazil. The individuals from *C. cochlearius* and *E. ruber* were carefully attended to by the veterinarians overseeing Parque Zoológico Mangal das Garças, while the *S. sibilatrix* individual was euthanized in accordance with the ethics committee mentioned. The euthanasia was necessary because the individual was found injured, for which recovery was unfortunately not possible. The euthanasia method used for the *S. sibilatrix* individual was a lethal dose of Ketamine 5% (300mg/kg)/ Xylazine 2% (50 mg/kg), administered intravenously. A small skin biopsy, collected from each individual using a surgical punch, were used to establish fibroblast cell cultures according to Furo et al. [22]. Chromosomes were obtained by standard methodology, using colcemid (Gibco) for metaphase arrest (1 hour incubation), hypotonic treatment with 0.075 M KCl for 15 minutes, and cell fixation in methanol/acetic acid (3:1) and kept in freezer for further experiments.

### Fluorescence *in situ* hybridization (FISH)

FISH experiments were performed using whole-chromosome probes of *G. gallus* (GGA 1–10) generated by flow-sorting at Cambridge Resource Centre for Comparative Genomics (Cambridge, UK). The probes were labeled with biotin (Roche Diagnostics, Mannheim, Germany) or digoxigenin (Roche Diagnostics, Mannheim, Germany) by degenerate oligonucleotide-primed polymerase chain reaction (DOP-PCR) and detected with streptavidin-CY3 (Invitrogen, Waltham, EUA) or sheep anti-digoxigenin FITC (Invitrogen, Waltham, EUA), respectively. Protocols for hybridization, stringency washes and detection were as described previously by Kretschmer et al. [23]. Chromosomes were counterstained with an antifade solution containing 4′,6-diamidino-2-phenylindole (DAPI) (Sigma-Aldrich, St. Louis, MO, USA). For each probe, at least 10 metaphase spreads per individual were analyzed to confirm FISH results using a Zeiss Axioplan2 fluorescent microscope and Axionvisio 4.8 software (Zeiss, Jena, Germany).

*G. gallus* probes were employed due to their similarity in karyotype to the ancestral avian lineage (Palaeognathae) [12, 13], making them a reference in cytogenetics and genetics studies. Besides that, the homology of chromosomes of different Galliformes species was one of the first and a classic example of the role of whole chromosome fusions in the avian karyotype evolution and systematics [24].

## Results

### Karyotype description

*C. cochlearius* has a karyotype composed of 74 chromosomes, with 11 pairs of macrochromosomes, including the sex chromosomes, and 26 pairs of microchromosomes. The seventh and tenth pairs are metacentric, the first, second, third, fourth, fifth, and sixth are submetacentric, and the pairs eighth and nineth are telocentric. The Z chromosome is submetacentric (Fig 1A and 1B). The W chromosome was not investigated because we analyzed a male individual.

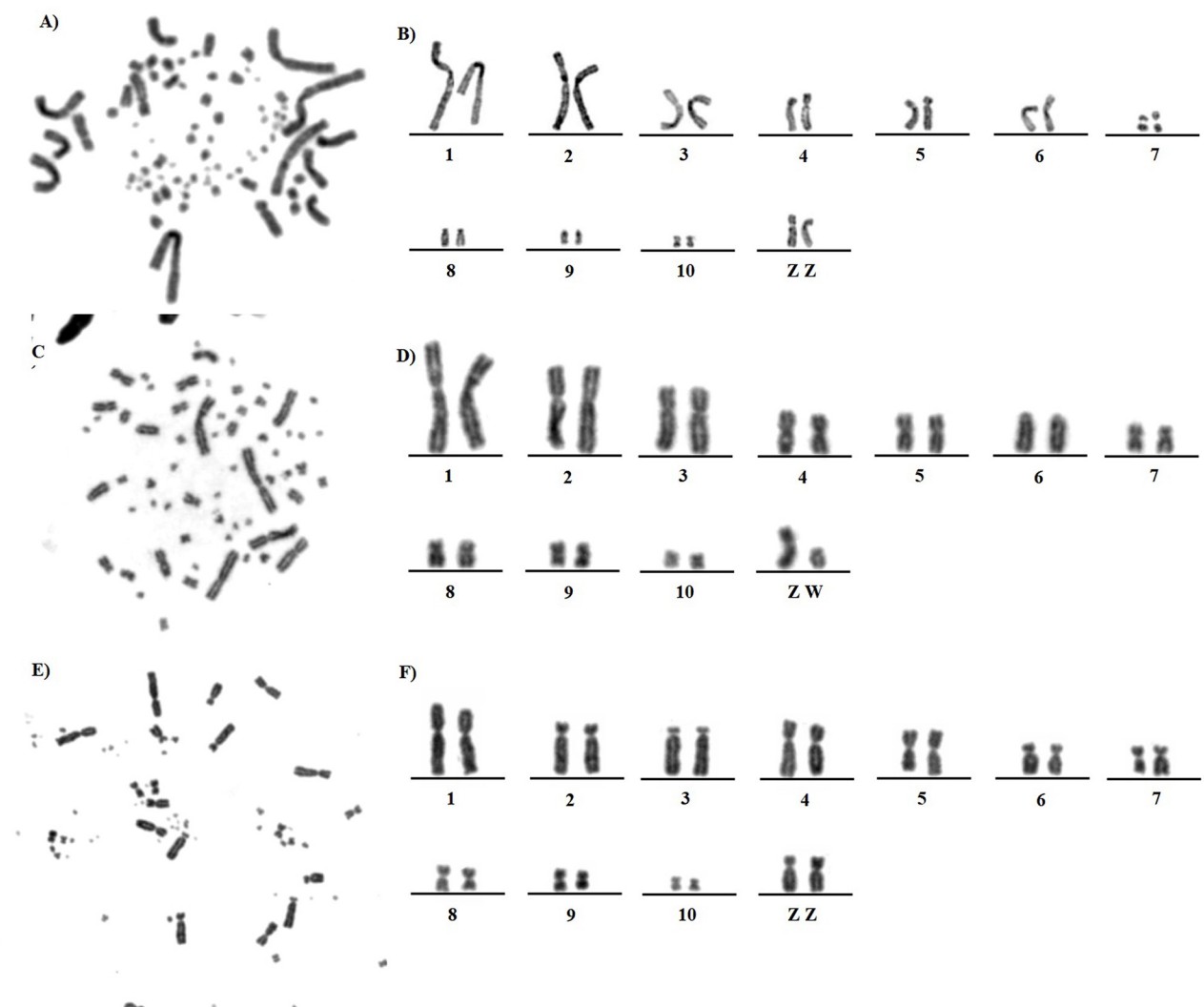

**Fig 1. Metaphase and partial karyotype of a *C. cochlearius* (A, B), *S. sibilatrix* (C, D), and *E. ruber* (E, F).**

*S. sibilatrix* has a karyotype composed of 62 chromosomes, with 11 pairs of macrochromosomes, including the sex chromosomes, and 20 pairs of microchromosomes. The fourth, fifth, seventh, nineth, and tenth are metacentric, the first, second, third, and eighth are submetacentric, and the sixth is telocentric. The Z chromosome is metacentric, and the W is acrocentric (Fig 1C and 1D).

The karyotype of *E. ruber* includes 66 chromosomes, with 10 pairs of macrochromosomes, including the sex chromosomes, and 23 pairs of microchromosomes. The fifth, eighth, and nineth pairs are metacentric, the first and fourth pairs are submetacentric, and the second, third, sixth, and seventh are acrocentric. The Z sex chromosomes is submetacentric (Fig 1E and 1F).

### Fluorescence *in situ* hybridization (FISH)

In *C. cochlearius* (CCO), the *G. gallus* probes (GGA 1–10) produced signals in 11 chromosomes (Fig 2). GGA1, GGA2, GGA3, GGA4, GGA5, GGA6, GGA7, GGA8, GGA9 and

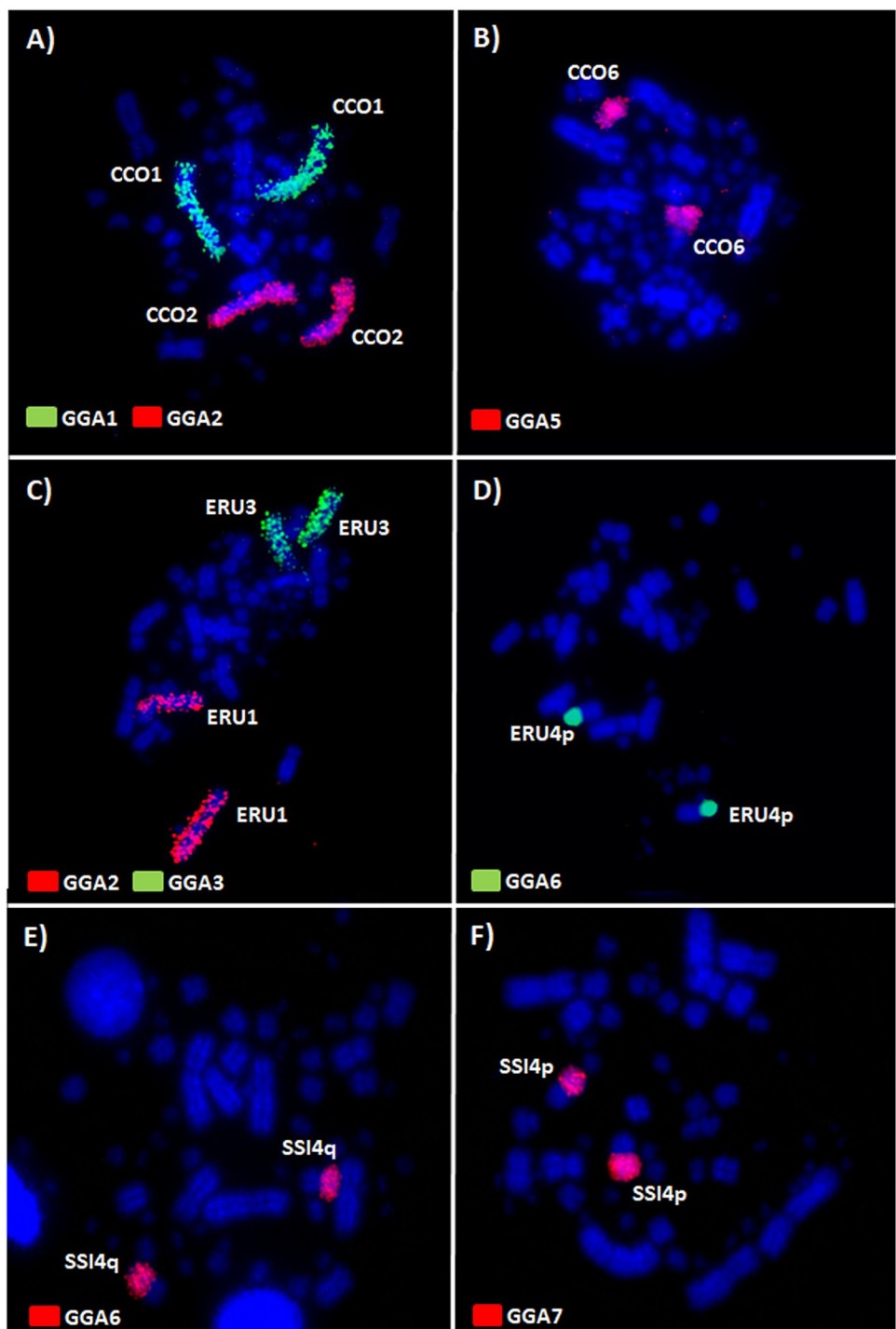

**Fig 2. Representative FISH of chicken (GGA) macrochromosome paints on DAPI-stained chromosomes of *C. cochlearius* (CCO), *S. sibilatrix* (SSI), and *E. ruber* (ERU).** The chromosome probes used are indicated on the left bottom, biotin-CY3 (red) and digoxigenin-FITC (green).

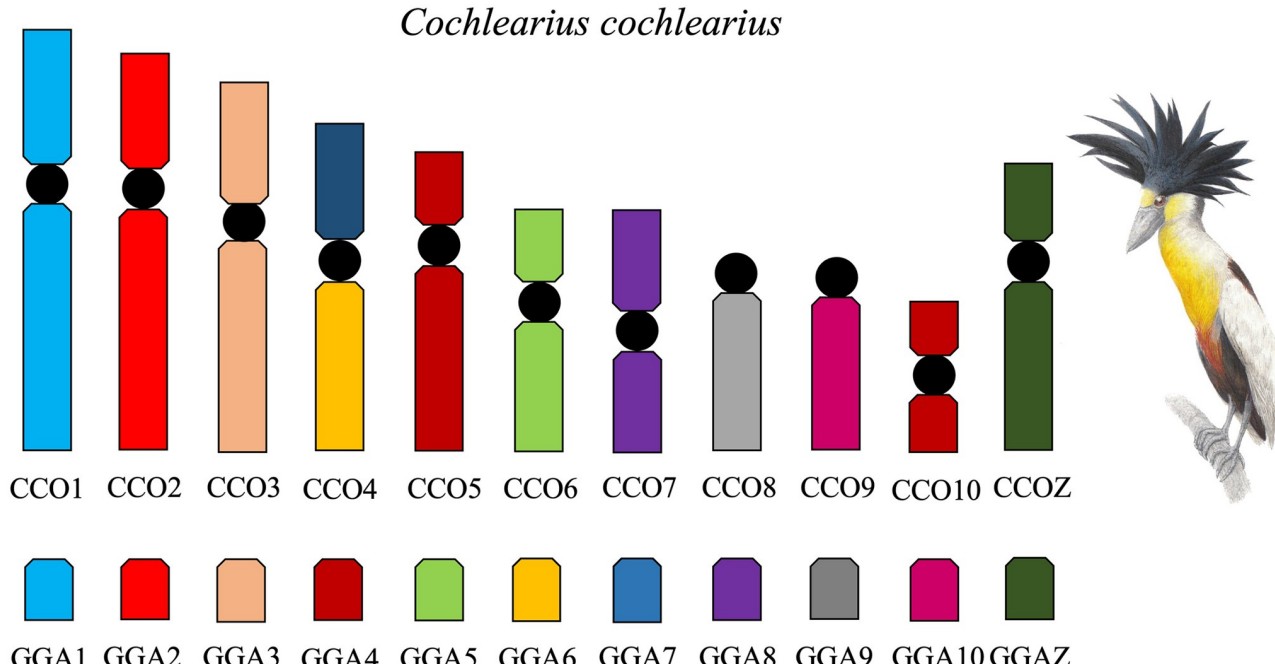

**Fig 3. Homology map between _C. cochlearius_ (CCO) and _G. gallus_ (GGA) as detected by fluorescence _in situ_ hybridization (FISH) using _G. gallus_ probes.**

GGA10 hybridized, respectively, to CCO1, CCO2, CCO3, CCO5 and CCO10, CCO6, CCO4q and CCO4p, CCO7, CCO8, and CCO9. The comparative map of _G. gallus_ probes onto _C. cochlearius_ is shown in Fig 3.

The _G. gallus_ probes produced signals in 11 pairs in _S. sibilatrix_ (SSI) (Fig 2). GGA1, GGA2, GGA3, GGA4, GGA5, GGA6, GGA7, GGA8, GGA9, and GGA10 hybridized, respectively, to SSI1, SSI2, SSI3, SSI5 and SSI10, SSI6, SSI4q, SSI4p, SSI7q, SSI8q, and SSI9q. Fig 4 shows the homology between _G. gallus_ and _S. sibilatrix_.

_G. gallus_ probes (GGA 1–9) produced signals in 11 chromosomes in _E. ruber_ (ERU) (Fig 2). GGA1, GGA2, GGA3, GGA4, GGA5, GGA6, GGA7, GGA8, and GGA9 hybridized, respectively, to ERU2 and ERU4q, ERU1, ERU3, ERU6 and ERU9, ERU7, ERU4p, ERU5q, ERU5p, and ERU8. The homology map between _G. gallus_ probes and _E. ruber_ is presented in Fig 5.

## Discussion

Although most birds have a karyotype consisting of 10 pairs of macrochromosomes and 30 pairs of microchromosomes [11–13], some orders of birds show a deviation of this typical avian karyotype. As an example, Pelecaniformes members have a diploid number ranging from 2n = 50 in _P. falcinellus_ [9] to 2n = 74 in _C. cochlearius_ [10]. In this study, we examined the chromosome organization of three Pelecaniformes species with varying diploid numbers, namely _C. cochlearius_ (CCO, 2n = 74), _S. sibilatrix_ (SSI, 2n = 62), and _E. ruber_ (ERU, 2n = 66). Our findings suggest that interchromosomal rearrangements, particularly fusion events involving macro and microchromosomes, play a fundamental role in the chromosome evolution of these species.

Among the five families within Pelecaniformes, cytogenetics studies in Scopidae, Balaenicipitidae, and Pelecanidae members, have been based solely on analyses using conventional

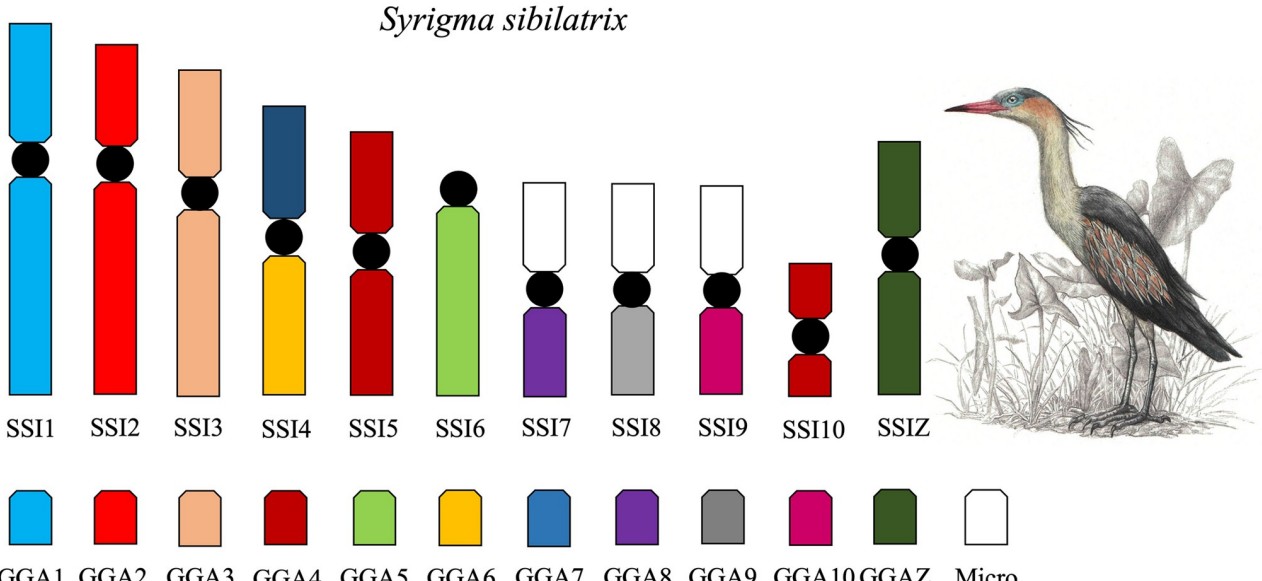

**Fig 4. Homology map between *S. sibilatrix* (SSI) and *G. gallus* (GGA) as detected by fluorescence *in situ* hybridization (FISH) using *G. gallus* probes.** Micro = Microchromosome.

staining [10, 25–27]. These studies indicated a low diploid number in these families, ranging from 2n = 66 to 72, similar to the species studied here and by Wang et al. [17], that reinforce the decreasing diploid numbers during the evolution of this group, considering that the putative avian ancestral karyotype has 80 chromosomes [12].

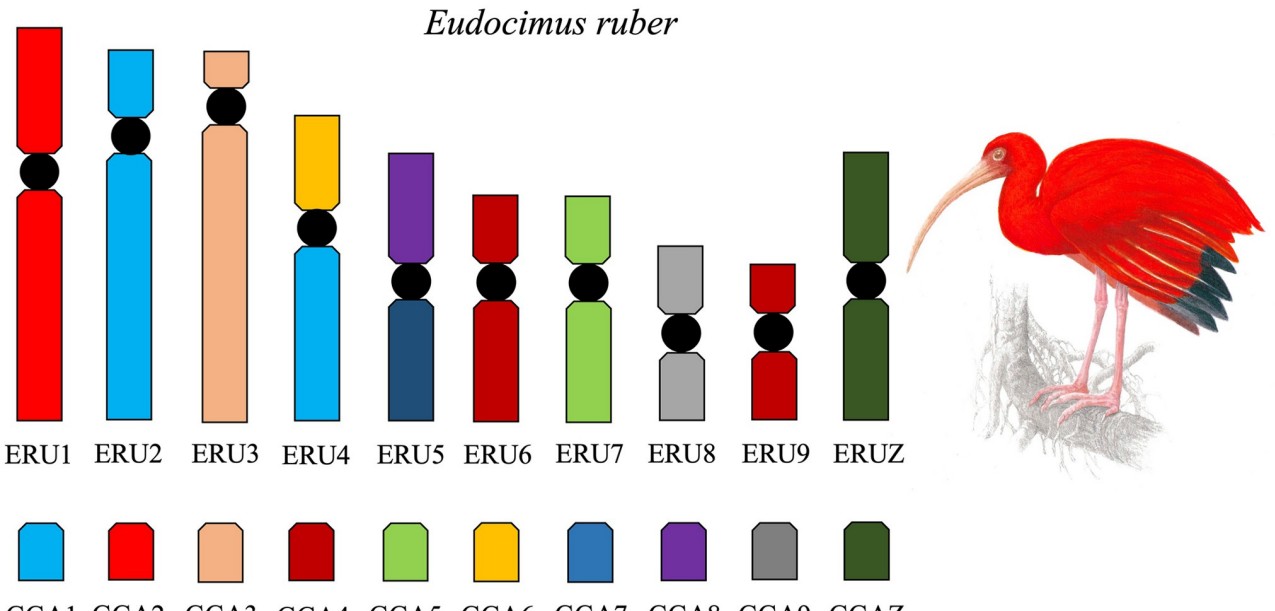

**Fig 5. Homology map between *E. ruber* (ERU) and *G. gallus* (GGA) as detected by fluorescence *in situ* hybridization (FISH) using *G. gallus* probes.**

The FISH experiments described here show a distinct pattern in the two Ardeidae species. In *C. cochlearius* (2n = 74), most of *G. gallus* probes (GGA 1–10) are conserved, as in the putative avian ancestor. Only the fusion between GGA6 and GGA7 (CCO4) was found in this species. Considering that the putative avian ancestral karyotype has 80 chromosomes [12], two additional fusions involving microchromosomes must have occurred to explain the diploid number of 74 in *C. cochlearius*. The few chromosome rearrangements detected in *C. cochlearius* provides further evidence supporting its phylogenetically basal position within the Ardeidae family [20]. In *S. sibilatrix* (2n = 62) we also detected the fusion between the GGA6 and GGA7 (SSI4), but in addition, we found evidence of microchromosome fusions, as none of the probes used in our study hybridized to the short arms of chromosome pairs SSI7, SSI8, and SSI9. Hence, six additional microchromosome fusions are necessary to explain the diploid number of 62 in *S. sibilatrix*. Therefore, in addition to being considered a derived species within the Ardeidae family [21], *S. sibilatrix* exhibits a more reorganized karyotype, further supporting its phylogenetic position. Recently, a study analyzing two Ardeidae (*A. cinerea* and *E. garzetta*) also indicate the occurrence of fusions of microchromosomes and macrochromosomes [17]. The same hybridization pattern was found in *A. cinerea* and *E. garzetta*, which six fusions were recognizable, including the GGA7/GGA8. Our study indicated different hybridization pattern in *C. cochlearius* and in *S. sibilatrix*, and none of them was similar to *A. cinerea* and *E. garzetta*. Hence, our study shows that different chromosome rearrangements occur in different Ardeidae species.

In contrast to the Ardeidae species investigated here, *E. ruber* (2n = 66), representing the Threskiornithidae family, showed a fusion between GGA7 and GGA8 in ERU5, and a centric fission in GGA1 producing rearrangements in ERU2 and ERU4. However, after the fission, the resulting chromosome homologous to the GGA1p underwent different rearrangements in each species. In *E. ruber*, it is found fused to homologous GGA6, while in *N. nippon* the association GGA1p/GGA9 is observed [17]. Furthermore, six additional microchromosome fusions are necessary to explain the decreased diploid number of 66 in *E. ruber*.

Interestingly, fusions have been described by chromosome painting in other species from this family [17]. In addition, Wang et al. [17] proposed that the association GGA7/GGA8 is a potential cytogenetic signature that unites the Ardeidae and Threskiornithidae families. However, in *C. cochlearius* and *S. sibilatrix*, the homologous chromosome to GGA 7 involves a fusion event with GGA6. Thus, the comparison of our results with the data from Wang et al. [17] indicate that fusions GGA7/8 and GGA7/6 are not chromosome signatures in the Pelecaniformes. In fact, these fusions have been found in several unrelated avian lineages, such as Columbiformes, Charadriiformes, Psittaciformes, and Gruiformes [11, 13], indicating that they are examples of convergent chromosome evolution.

With regard to microchromosomes, previous studies have indicated that these tiny elements have been maintained stably during karyotype evolution in most birds, with some exceptions, e.g., in Ciconiiformes, Cuculiformes, Falconiformes, Pelecaniformes, Psittaciformes, and Suliformes [17–19, 28–31]. Here we establish that microchromosome fusions play an important role in Pelecaniformes chromosome evolution. Interestingly, Ciconiiformes and Suliformes, two closely related avian orders to Pelecaniformes [8], also have extensive fusions involving microchromosomes, suggesting that these rearrangements could be present in their common ancestor. It has been proposed that breakpoint regions involved in chromosomal rearrangements are usually associated with genomic features, including transposable elements, and conserved noncoding sequences [32, 33]. Hence, further studies may reveal chromosomal signatures of these orders that indicate a genomic reason why these species have in common atypical karyotype.

To date, *C. cochlearius* has been found to have the highest diploid number among species of Pelecaniformes, with 2n = 74. Hence, our data corroborate the basal phylogenetic position of this species, and we propose that it has the most plesiomorphic karyotype in the order. In support of this hypothesis, *C. cochlearius* also has less microchromosome fusions than other species in the order. Using cytochrome b and DNA hybridization it is not clear whether *Cochlearius* or *Tigrisoma* are at the basal end of Ardeidae tree [20]. However, among the species of *Tigrisoma*, only *Tigrisoma lineatum* has a characterized karyotype, with 2n = 72 [30]. Hence, considering all available data, we suggest that *C. cochlearius* is at the basal end of Ardeidae tree.

## Conclusions

In this study we compared the karyotype of three Pelecaniformes species, using FISH experiments with whole chromosome probes of *G. gallus*. The results indicated that their karyotypical divergence included the occurrence of fissions of macrochromosomes and fusions of both macrochromosomes and microchromosomes. These types of rearrangement have also been observed in other closely related orders, namely Ciconiiformes and Suliformes. However, no common cytogenetic signature has been found among the species from these orders. Considering these orders are phylogenetically close, and the stability of avian syntenic groups, especially the microchromosomes in most groups of birds, we speculate that Pelecaniformes, Ciconiiformes and Suliformes may have some genome organization features that facilitated the occurrence of interchromosomal rearrangements involving microchromosomes. Further studies should be addressed to investigate this hypothesis.

## Acknowledgments

Authors would like to thank all colleagues from the "Grupo de Pesquisa Diversidade Genética Animal" from Universidade Federal do Pampa, the "Laboratório de Cultura de Tecidos e Cito-genética" from Instituto Evandro Chagas (Ananindeua, PA, Brazil), and PROPESP-UFPA for technical and institutional support. We are grateful to Alex Pinheiro de Araújo for the illustration of the *Cochlearius cochlearius*, *Syrigma sibilatrix* and *Eudocimus ruber* used in Figs 3–5, respectively.

## Author Contributions

**Conceptualization:** Igor Chamon Assumpção Seligmann, Malcolm Ferguson-Smith, Rafael Kretschmer, Edivaldo Herculano C. de Oliveira.

**Data curation:** Michelly da Silva dos Santos, Rafael Kretschmer.

**Formal analysis:** Igor Chamon Assumpção Seligmann, Fabio Augusto Oliveira Silva, Rafael Kretschmer.

**Funding acquisition:** Ricardo José Gunski, Analía del Valle Garnero, Malcolm Ferguson-Smith.

**Investigation:** Ivanete de Oliveira Furo, Michelly da Silva dos Santos, Ricardo José Gunski.

**Methodology:** Igor Chamon Assumpção Seligmann, Ivanete de Oliveira Furo, Michelly da Silva dos Santos, Ricardo José Gunski, Analía del Valle Garnero, Fabio Augusto Oliveira Silva, Patricia O´Brien, Rafael Kretschmer.

**Project administration:** Rafael Kretschmer.

**Resources:** Edivaldo Herculano C. de Oliveira.

**Supervision:** Patricia O´Brien, Rafael Kretschmer, Edivaldo Herculano C. de Oliveira.

**Validation:** Igor Chamon Assumpção Seligmann, Michelly da Silva dos Santos, Edivaldo Herculano C. de Oliveira.

**Visualization:** Igor Chamon Assumpção Seligmann, Michelly da Silva dos Santos.

**Writing – original draft:** Igor Chamon Assumpção Seligmann, Patricia O´Brien, Malcolm Ferguson-Smith, Edivaldo Herculano C. de Oliveira.

**Writing – review & editing:** Ivanete de Oliveira Furo, Ricardo José Gunski, Analía del Valle Garnero, Patricia O´Brien, Malcolm Ferguson-Smith, Edivaldo Herculano C. de Oliveira.

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
