## [Decision Letter · Decision Letter 0]

4 Oct 2023

PONE-D-23-23651Comparative Chromosome Painting in Three Pelecaniformes Species (Aves): Exploring the Role of Macro and Microchromosome Fusions in Karyotypic EvolutionPLOS ONE

Dear Dr. de Oliveira,

Thank you for submitting your manuscript to PLOS ONE. After careful consideration, we feel that it has merit but does not fully meet PLOS ONE’s publication criteria as it currently stands. Therefore, we invite you to submit a revised version of the manuscript that addresses the points raised during the review process.

We look forward to receiving your revised manuscript.

Kind regards,

Qinghua Shi

Academic Editor

PLOS ONE

2. In your Methods, please provide full details of the handling of  C. cochlearius, E. ruber and S. sibilatrix during the experiment. Also provide the details of methods used to obtain the biological samples used in your analyses.

“This research was funded by Conselho Nacional de Desenvolvimento Científico e Tecnológico (CNPq, Proc. 407285/2021-0 to Analía del Valle Garnero, Proc. 304781/2022-3 to Edivaldo Herculano C. de Oliveira).”

6. We note that Figures 3,4 and 5 in your submission contain copyrighted images. All PLOS content is published under the Creative Commons Attribution License (CC BY 4.0), which means that the manuscript, images, and Supporting Information files will be freely available online, and any third party is permitted to access, download, copy, distribute, and use these materials in any way, even commercially, with proper attribution. For more information, see our copyright guidelines: http://journals.plos.org/plosone/s/licenses-and-copyright.

a. You may seek permission from the original copyright holder of Figures 3,4 and 5 to publish the content specifically under the CC BY 4.0 license.

Reviewers' comments:

Reviewer's Responses to Questions

**Comments to the Author**

1. Is the manuscript technically sound, and do the data support the conclusions?

Reviewer #1: Yes

2. Has the statistical analysis been performed appropriately and rigorously? 

Reviewer #1: N/A

3. Have the authors made all data underlying the findings in their manuscript fully available?

Reviewer #1: Yes

4. Is the manuscript presented in an intelligible fashion and written in standard English?

Reviewer #1: Yes

5. Review Comments to the Author

Reviewer #1: The article Comparative Chromosome Painting in Three Pelecaniformes Species (Birds): Exploring the Role of Macro and Microchromosome Fusions in Karyotypic Evolution by Seligmann et al. presents interesting data on the chromosomal evolution of the order Pelecaniformes represented by three species from two different families. The results are innovative concerning comparisons with chromosomal probes, showing that chromosomal rearrangements are related to species differentiation. Two points should be analyzed: in the introduction, the authors mention that the species were selected because C. cochlearius is considered phylogenetically a basal species in the family, and S. sibilatrix is more derived, which is a very interesting fact about evolution. However, this fact should have been considered throughout the article concerning what was found. Another point is that the authors should review the text describing the karyotypes, especially the number of macrochromosomes in Figure 1, as they disagree.

6. PLOS authors have the option to publish the peer review history of their article (what does this mean?). If published, this will include your full peer review and any attached files.

Reviewer #1: No

---

## [Author Response · Author response to Decision Letter 0]

7 Nov 2023

Dear Reviewer, 

First of all, we would like to thank you for taking your time, effort and patience in order to try to make the manuscript better. We followed your suggestions and answer your doubts. We hope we have achieved these goals.

Sincerely

The authors

Reviewer 1

The article Comparative Chromosome Painting in Three Pelecaniformes Species (Birds): Exploring the Role of Macro and Microchromosome Fusions in Karyotypic Evolution by Seligmann et al. presents interesting data on the chromosomal evolution of the order Pelecaniformes represented by three species from two different families. The results are innovative concerning comparisons with chromosomal probes, showing that chromosomal rearrangements are related to species differentiation. 

Authors: Thank you very much for your positive feedback.

Two points should be analyzed: in the introduction, the authors mention that the species were selected because C. cochlearius is considered phylogenetically a basal species in the family, and S. sibilatrix is more derived, which is a very interesting fact about evolution. However, this fact should have been considered throughout the article concerning what was found. 

Authors: We have discussed these results at lines 234, 235, 239-241. We also have already discussed the basal position of C. cochlearius in the last paragraph in the introduction.

Another point is that the authors should review the text describing the karyotypes, especially the number of macrochromosomes in Figure 1, as they disagree.

Authors: We have reviewed the text and the Figure 1. The number of macrochromosomes are correct in the text and in the figure, once we considered 11 pairs of macrochromosomes including the sex chromosomes. We have made it clear in the manuscript.

---

## [Editor Report · Decision Letter 1]

9 Nov 2023

Comparative chromosome painting in three pelecaniformes species (Aves): Exploring the role of macro and microchromosome fusions in karyotypic evolution

PONE-D-23-23651R1

Dear Dr. Edivaldo Herculano Corrêa de Oliveira,

We’re pleased to inform you that your manuscript has been judged scientifically suitable for publication and will be formally accepted for publication once it meets all outstanding technical requirements.

Kind regards,

Qinghua Shi

Academic Editor

PLOS ONE

Additional Editor Comments (optional):

his manuscript has been revised based on the suggestions. I am happy with the revision and thus think it can be accepted for publication.
---

## [Editor Report · Acceptance letter]

14 Nov 2023

PONE-D-23-23651R1 

Comparative chromosome painting in three Pelecaniformes species (Aves): Exploring the role of macro and microchromosome fusions in karyotypic evolution 

Dear Dr. de Oliveira:

I'm pleased to inform you that your manuscript has been deemed suitable for publication in PLOS ONE. Congratulations! Your manuscript is now with our production department. 

Kind regards, 

on behalf of

Professor Qinghua Shi 

Academic Editor

PLOS ONE